# PowerNet: Truncated Matrix Power Series as Quasi-Equivariant Layers

## Abstract

Despite being theoretically well-grounded, enforcing strict equivariance in deep learning models has shown to be harmful in some cases. The problem is that most available data does not follow mathematically precise rules, is noisy, and is not strictly group-structured. While soft equivariance approaches attempt to address these issues, they often struggle to maintain group structure and lack strong theoretical guarantees, potentially compromising the benefits of equivariance. Here we introduce the concept of *quasi equivariances*, where group structure is maintained but the associated parameters become distributions, and implement it in the proposed *PowerNet* architecture. Similar to CNNs, PowerNet is constructed by interlacing truncated matrix power series with non-linearities. We show how the base matrix used to define the power series can instill quasi-equivariance in a natural way. Finally, we provide results for augmented MNIST classification and transformation magnitude regression in addition to classification of CIFAR-10.

## 1 Introduction

Although group equivariant convolutional networks (Cohen & Welling, 2016) can be limitlessly expressive (Yarotsky, 2018), enforcing strict equivariance in neural networks has shown to be more limiting than beneficial (Liu et al., 2018; Wang et al., 2022). Real-world data and tasks rarely exhibit group-structured properties. Due to the closure property, group actions form inherently strict algebraic structures and most datasets encountered in practice are not naturally distributed according to known groups.

An exception can be made for tasks that are inherently mathematical or group structured, such as molecules (Gilmer et al., 2017), PDE data (Brandstetter et al., 2023), dynamical systems (Yang et al., 2024), or neural networks themselves (Navon et al., 2023; Kofinas et al., 2024). A typical approach is to assume an underlying group structure and parameterize a distribution of the magnitudes of the transformation on the orbits of the associated one-parameter group (Falorsi et al., 2019; Dehmamy et al., 2021; van der Ouderaa & van der Wilk, 2022; Gabel et al., 2023). Mathematical results in group theory are usually only known for compact or linear groups, the contrary of the latter taking us beyond traditional (linear) representation theory, usually used in geometric deep learning. A natural extension of this observation is that groups are idealized abstractions of transformations, which data is then assumed to conform to, but often does not.

In this paper, we propose to revisit the recently proposed *soft equivariance* concept from a weight sharing perspective. The connection between weight sharing and equivariance has been known since the introduction of CNNs (LeCun, 1989; Ravanbakhsh et al., 2017; Maron et al., 2019). Soft equivariance approaches aim to relax strict symmetry constraints, allowing models to handle real-world data that often doesn't perfectly adhere to mathematical symmetries. However, these methods often struggle to maintain the underlying group structure and lack strong theoretical guarantees, potentially compromising the benefits of equivariance while introducing additional computational complexity. The motivation for studying a relaxed version of group equivariance, which we will refer to as *quasi-equivariance*, is threefold. In mathematical settings, symmetry transformations of a given task can often be described precisely by groups, whether as continuous (e.g., rotations) or discrete group actions. However, real-world data is noisy, imprecise, and seldom exhibits such exact group symmetries. Quasi-equivariance allows models to capture underlying patterns without committing to strict group-theoretic assumptions. Second, like soft equivariance, quasi-equivariance

enables models to remain sensitive to probabilistic magnitudes of transformations, making the network more adaptive to varying magnitudes of transformation in data. In real-world settings, where robustness w.r.t. small deformations is often a sought-after property, this property can lead to vast improvement in out-of-distribution performance. Third, learnable inductive biases or structural priors: By exploring equivariance through the lens of weight sharing, the model can learn inductive biases more naturally.

To illustrate our approach, we draw an analogy between Convolutional Neural Networks (CNNs) and Graph Neural Networks (GNNs). Both models employ weight-sharing principles derived from symmetry assumptions, i.e., shift and permutation matrices respectively. With *PowerNet*, we propose a generalization for any base matrix, which enables more flexible structure learning. This generalization does not inherently recover a basic multi-layer perceptron (MLP), however. To evaluate the effectiveness of our model, we report augmented MNIST classification and regression on the magnitude of the parameters. Additionally, we ran our model on CIFAR-10 for classification.

This approach has promising implications for future research, particularly when combined with continual learning. A neural network equipped with learnable equivariances could dynamically update its connectivity scheme as it processes new data, allowing it to adaptively learn new inductive biases over time. Such a system could evolve its internal structure in response to experience, potentially abstracting over consecutive inductive biases, whether in series (this work) or in parallel. The latter of which we leave as a possibility for future work.

**Contributions**  With this investigation of the PowerNet architecture, we introduce a number of contributions to the field of geometric deep learning:

- A novel way of interpreting equivariant neural networks by relaxing the strict constraint group actions place on the traditional group equivariant architectures considered in the field,
- a connection between Laurent polynomials, weight sharing, and equivariant neural networks that could help with structural bias learning,
- the PowerNet architecture and library, which is fast, parameter efficient, open source, and easy to use.

**Limitations**  Besides the considerable low parameter count PowerNet is able to perform tasks such as augmented MNIST classification and CIFAR with, it does not currently beat state-of-the-art on the latter.

**Reproducibility**  We provide all scripts and settings used for the experiments contained in this paper. The PowerNet library will be made open-source in order to allow for further experimentation and incorporation of the methods presented here.

## 2  BACKGROUND AND RELATED WORK

This work is connected to various subareas in geometric deep learning. Therefore, in this section, we will sketch the context of this paper, how it relates to previous results, and how it differs.

### 2.1  GROUPS, GRAPHS, AND SELF-ATTENTION

The operation of sharing weights in deep neural networks has been an intense area of study (Ravanbakhsh et al., 2017; Maron et al., 2019). It ranges from exploiting the symmetries of the weights in neural networks in order to improve training Neural Fields (Navon et al., 2023; Kofinas et al., 2024) to performing lifting operations in group convolutions that make the kernel mimic the group transformation of interest such that the model becomes equivariant (Cohen & Welling, 2016; Kondor & Trivedi, 2018; Bekkers et al., 2018; Weiler & Cesa, 2019). The connection between CNNs, group convolutions, and GNNs has been discussed in great detail (Bronstein et al., 2021). In a closely related work, matrix functions have been proposed for applications to graph-structured data (Batatia et al., 2024).

The success of transformer-based models has attracted interest from the geometric deep learning field. Group equivariant attention has been shown to be possible when applied to the positional en-

codings (Romero & Cordonnier, 2021) and the relationship to wavelets in time-series for translation and scale equivariance is also noteworthy (Romero et al., 2024).

## 2.2 INDUCTIVE BIAS LEARNING

In the context of inductive bias learning, the irrelevancies of a given task, usually formalized mathematically as symmetry groups Knigge et al. (2022), can be exploited by geometric methods that either lead to weight-sharing schemes within the neural network Zhou et al. (2020); Finzi et al. (2021); van der Ouderaa et al. (2023) or overcompensate with additional computational operations for when the input undergoes the expected transformation Kondor & Trivedi (2018); Bekkers et al. (2018); Romero & Lohit (2022).

## 3 THE *PowerNet* ARCHITECTURE

We take inspiration from CNNs, in which convolutions can be rewritten as matrix multiplications of a circulant matrix with the flattened, i.e., vectorized, image. First, we define the PowerLayer and PowerBlock. Mathematically, we are interested in Laurent polynomials over the reals, a slight variation on the usual group-based exposition given in most works (Maron et al., 2019; Bronstein et al., 2021). The variable is a matrix that we will refer to as the base matrix.

### 3.1 LAURENT POLYNOMIALS AND LAURENT CONVOLUTION

**Definition** A *Laurent polynomial* $\mathcal{P} \in \Pi(z, z^{-1})$ is a polynomial of positive and negative integer powers of a variable $z$ over a field $\mathbb{F}$. Formally,

$$\mathcal{P} = \sum_{i \in \mathbb{Z}} \theta_i z^i,$$

for a finite number of non-zero $\theta_i \in \mathbb{F}$.

Consider parametrizing the weight matrix of a neural network as a Laurent polynomial of a square *base matrix* $\boldsymbol{M} \in \mathbb{R}^{d \times d}$ over $\mathbb{R}$ in order to generalize the concept of a convolutional layer. Note, in this case, $\mathcal{P} = \boldsymbol{P} \in \Pi\left(\boldsymbol{M}, \boldsymbol{M}^{-1}\right) \subset \mathbb{R}^{d \times d}$. In other words, we are interested in weight matrices that can be written as a truncated power series of a matrix. We will refer to the linear mapping $\boldsymbol{P}\boldsymbol{x}$ as a *Laurent convolution*.

Hence, we define a single *PowerLayer* as $f_\theta(\boldsymbol{x}|\boldsymbol{M}) = \sigma(\boldsymbol{P}\boldsymbol{x} + \boldsymbol{b})$, with feature vector and bias term $\boldsymbol{x}, \boldsymbol{b} \in \mathbb{R}^d$. The non-linear activation function is denoted by $\sigma$ and the resulting feature vector is of the same size as the input to the layer, namely $f_\theta(\boldsymbol{x}|\boldsymbol{M}) = \boldsymbol{y} \in \mathbb{R}^d$. This parametrization of the weight matrix as a Laurent convolution allows for a natural definition of kernel size, dilation, and convolutions. In particular, if we let the powers range from $-K$ to $K$, the number of parameters is $2K + 1 \sim O(K)$ where $K \in \mathbb{N}_0 = \{1, 2, 3, ...\}$. This gives PowerLayers a low parameter count, relative to most architectures.

Dilation of the kernel can easily be controlled: By analogy with CNNs, we simply multiply the powers in the Laurent series by $D \in \mathbb{N}_0$. Locality is easily relaxed by choosing a different base matrix. Note that the layer still performs an affine transformation of the feature vector $\boldsymbol{x}$.

### 3.2 GENERAL FORMULATION

For input channel $c_i$, the feature map $\boldsymbol{x}^{(c_i)} \in \mathbb{R}^d$ is acted upon by a generalized PowerLayer as follows:

$$f_\theta^{(c_o)}(\boldsymbol{x}|\boldsymbol{A}\boldsymbol{B}...\boldsymbol{Z}) = \sigma\left[\left(\sum_{i,j,...,q \in \mathcal{K}} \theta_{ij...q}^{(c_o,c_i)} \boldsymbol{A}^i \boldsymbol{B}^j ... \boldsymbol{Z}^q\right) \boldsymbol{x}^{(c_i)} + \boldsymbol{b}\right], \quad \boldsymbol{A}, \boldsymbol{B}...\boldsymbol{Z} \in \mathbb{R}^{d \times d}, \boldsymbol{b} \in \mathbb{R}^d.$$

This defines the output feature map $\boldsymbol{y}^{(c_o)} \in \mathbb{R}^d$ for output channel $c_o$. The set of powers to select from is indicated by $\mathcal{K}$, $\theta_{ij...q}^{(c_o,c_i)}$ are the updatable parameters of the PowerLayer, and $\sigma$ is the non-linear activation function. If we wish, can fix the base matrices using Lie theory as follows: Let

the base matrix be the result of a matrix exponential such that $\boldsymbol{A} = e^{\boldsymbol{G}} = \sum \frac{1}{n!} \boldsymbol{G}^n$, where $\boldsymbol{G}$ is the generator of a Lie group transformation. When the action of the base matrices on the flattened image we recover equivariant layers. We can show that the weight matrix reflects the group action of a multi-parameter Lie group, i.e. $\boldsymbol{A}^i \boldsymbol{B}^j ... \boldsymbol{Z}^q = e^{iG_1} e^{jG_2} ... e^{qG_k}$, where the integers form the magnitudes of the transformation. According to Lie theory (Fulton & Harris, 1991), this product of exponentials is able to cover the component connected to the identity.

### 3.3 Lie Theory and the Shannon-Whitakker Interpolation

Clearly, we would like PowerNet to have the ability to encapsulate the usual group equivariance pervasive in the geometric deep learning literature. Even though the group assumption has been relaxed, here we show how PowerNet is still able to model group equivariance by choosing appropriate base matrices. The relevant kernels are the ones that mimic the group actions under consideration, this can be done by choosing a base matrix that permutes the pixels of the flattened image accordingly. Note that in most cases this will involve some aliasing effect. Since we are only interested in quasi-equivariance, aliasing effects should not worry us here. The key assumption is that depth, multiple kernels, pooling operations, and residual connections present in most deep learning models are sufficient to solve the task.

Using Lie theory, we can describe the problem as follows. The transformation of interest is defined by an element of a Lie algebra, and as long we can perform the exponential map, the result should be the group action we are interested in and a candidate for the base matrix of the PowerLayer. The elements of the Lie algebra can be obtained by first defining the derivative operator, a matrix that when exponetiated yields the shift matrix. One such approach involves using the *Shannon-Whitakker* (SW) *interpolation* to calculate the derivative operator, as was done in previous works attempting to parameterize Lie group in the context of machine learning (Rao & Ruderman, 1998; Dehmamy et al., 2021; Gabel et al., 2023).

To apply the operators to a grid, one must write the partial derivatives as matrices. Using the SW interpolation automatically assumes the function to be interpolated is periodic, although other interpolation schemes could have been chosen. We note that this scheme introduces some aliasing for transformations of low-resolution images, and forms one of the notable limitations of the current model. One could investigate other choices, such as bicubic interpolation, although deriving the differential operator for this scheme requires some additional analysis and is left for future work. Nevertheless, we pick this interpolation scheme for its ability to perform the transformations of interest using matrix-vector multiplication. Let $I$ be some real-valued signal. For a discrete set of $n$ points on the real line and $I(i + n) = I(i)$ for all samples $i$ from 1 to $n$, the SW interpolation reconstructs the signal for all $r \in \mathbb{R}$ as

$$I(r) = \sum_{i=0}^{n-1} I(i) Q(r - i),$$

$$Q(r) = \frac{1}{n} \left[ 1 + 2 \sum_{p=1}^{n/2-1} \cos\left(\frac{2\pi p r}{n}\right) \right]. \tag{1}$$

To obtain numerical expressions (matrices) for $\partial_x$, $Q$ can be differentiated with respect to its input. This then describes continuous changes in the one dimensional spatial coordinate at all $n$ points, i.e., $[\boldsymbol{D}_{\mathbb{R}}]_{ab} = \partial_a Q(a - b)$. The above can be extended to two dimensions by performing the Kronecker product of the result obtained for one dimension with the identity matrix, $\boldsymbol{D}_x = \boldsymbol{D}_{\mathbb{R}} \otimes \mathbb{I}$ and $\boldsymbol{D}_y = \mathbb{I} \otimes \boldsymbol{D}_{\mathbb{R}}$, mirroring the flattening operation applied to the input images. The parametrized generator for the 2D affine case, for example, looks like:

$$\boldsymbol{G}_\alpha = \sum_{i=1}^{6} \alpha_i \boldsymbol{D}_i, \tag{2}$$

where the $\boldsymbol{D}_i \in \mathbb{R}^{n^2 \times n^2}$ are the matrices that represent the operators $\partial_x, x\partial_x, y\partial_x, \partial_y, x\partial_y$, and $y\partial_y$, respectively. This can easily be extended to arbitrarily dimensional data by adding more factors to the above matrices, as was done above for the quadratic basis. One can see that performing this operation in pixel space scales poorly with signal length (or image width) $n$.

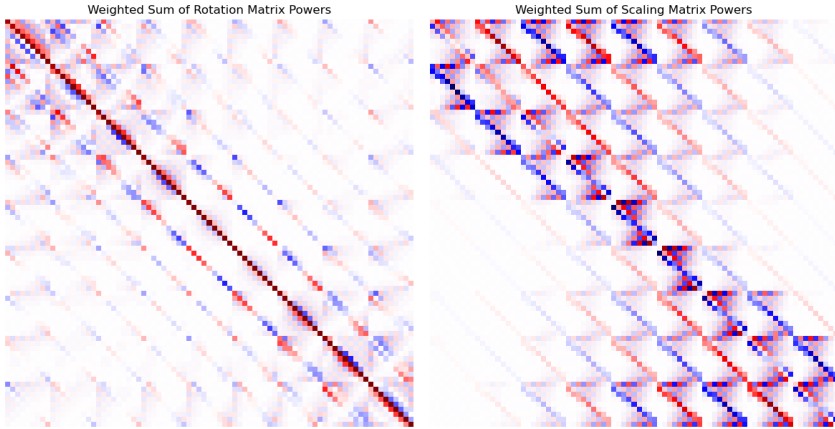

Figure 1: Examples of weight matrices with rotation (left) and scaling (right) quasi-equivariance for 9-by-9 flattened image inputs. The kernel values were sampled at random.

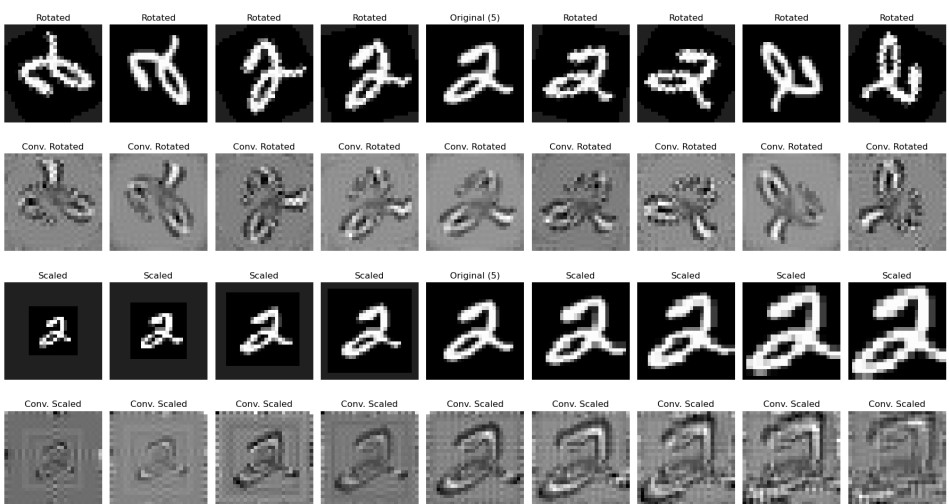

Figure 2: Applying the Laurent convolution on a sample MNIST digit for various rotation angles and scaling factors.

### 3.4 ONE-PARAMETER GROUPS

In the simplest version of a PowerNet layer, only one base matrix is chosen. This leads to the following formulation of a single *PowerLayer*:

$$f_\theta(\boldsymbol{x}|\boldsymbol{A}) = \sigma\left[\left(\sum_{i\in\mathcal{K}}\theta_i\boldsymbol{A}^i\right)\boldsymbol{x} + \boldsymbol{b}\right], \quad \boldsymbol{A}\in\mathbb{R}^{d\times d}, \boldsymbol{b}\in\mathbb{R}^d.$$

In this form, the model is almost identical to a group convolution (Cohen & Welling, 2016). The key difference is the emphasis on the truncated power series and the flexibility it provides in allowing for virtually any base matrix. Crucially, there is no further restriction on the layer, and it can be generalized to handle multiple channels in the usual way. Dilation can be introduced as follows: $\sum_{i\in K}\theta_i\boldsymbol{A}^{iD}$. (We refer the reader to Figure 1 and Figure 2 for some examples of how the weight matrix and Laurent convolution looks like for rotation and scaling.)

### 3.5 SPECIAL CASE: THE CNN

In the formalism introduced above, CNNs can be naturally described as Laurent convolutions using the shift matrices. For traditional, 2D convolutional layers, two such shift matrices need to be combined in order to define a $(2K_x + 1)$-by-$(2K_y + 1)$ sized kernel.

$$f_\theta(\boldsymbol{x}|\boldsymbol{S}_x\boldsymbol{S}_y) = \sigma\left[\left(\sum_{i,j\in K} \theta_{ij}\boldsymbol{S}_x^i\boldsymbol{S}_y^j\right)\boldsymbol{x} + \boldsymbol{b}\right], \quad \boldsymbol{S}_x, \boldsymbol{S}_y \in \mathbb{R}^{d\times d}, \boldsymbol{b} \in \mathbb{R}^d.$$

Adding rotation, one recovers a quasi-rototranslational neural network (Lafarge et al., 2021).

## 4 RESULTS

We show the results for applying PowerNet with various base matrices compared to the type of transformation that was used to obtain the augmented dataset.

### 4.1 EXPERIMENTAL SET-UP

With this work, we provide code for the `PowerNet` mini-library. This library allows a user to design custom quasi-equivariant deep neural networks for their own projects. The PowerNet architecture is characterized by the `PowerLayer()` class, which is defined by its `base_matrices`. Similar to CNNs, the user should provide kernel sizes (`powers`) in addition to the conventional `c_in` and `c_out` values, which correspond to the number of input and output channels of the PowerLayer, respectively. To make the construction of deep architectures easier, the `PowerBlock()` class provides options for the strength of residual connections (`rho`), multiple layers per block (`num_layers`), choice of non-linearities (`non_linearity`), batch or layer normalization, etc. (We refer the reader to Figure 3 for an example pipeline.)

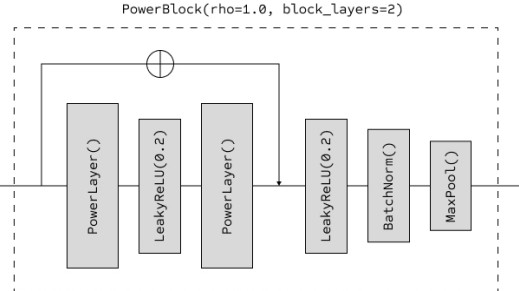

Figure 3: Example of a typical `PowerBlock()` pipeline used for the experiments in this work.

### 4.2 AUGMENTED MNIST REGRESSION AND CIFAR-10 CLASSIFICATION

In all experiments, `BatchNorm` (Ioffe & Szegedy, 2015), residual connections and `LeakyReLU(0.2)` was used since it mainly sped up training.

The experimental setup was as follows: 4 blocks, 2 layers per block, 32 channels, kernel size of 5, stride 1, with residual connection, batch size of 1024, average pooling, Adam optimizer (Kingma & Ba, 2015), learning rate 0.01, and weight decay 0.0001. The base matrices were chosen from rotation, scaling, and shift (in the "up" direction) and trained on rotated MNIST. The task was to predict the transformation magnitude, in this case, the rotation angle. The test performance (MSE) on the augmented rotation angle in radians was 1.485 (shift), 1.456 (scale), **0.949** (rotation) radians. The disparate result clearly shows the improvement when the correct inductive bias is used. This model only has 28k parameters.

For augmented 2xMNIST classification, namely a double sized image with the digit rotated (full range), scaled (between 0.8 and 1.2), and translated (max. 10 pixels) the setup was: 4 base matrices,

one for each transformation mentioned in the previous paragraph (shift up and shift right being two separate base matrices), 5 blocks, 2 layers per block, $[2, 4, 8, 16]$ channels, kernel size of 5, stride 1, max pooling, batch size of 64, Adam optimizer, learning rate 0.001, and weight decay 0.0001 the performance reached $84\%$ (identical performance to a CNN baseline with the same channel and max pooling structure). This model has 62k parameters.

Taking a larger model with 4 base matrices, 5 blocks, 1 layer per block, 32 channels, kernel size of 5, stride 1, max pooling, batch size of 64, Adam optimizer, learning rate 0.0001, and weight decay 0.0001 the performance reached $75\%$ for CIFAR-10 classification.

## 5 CONCLUSION

In this work, we introduced PowerNet, a neural network architecture that takes inspiration from CNNs and GNNs by parameterizing the weight matrix by a truncated matrix power series. Mathematically, we draw connections to Laurent polynomials and the duality between equivariant models and weight sharing. We show equal performance to baselines on an augmented MNIST dataset, and decent performance on CIFAR-10 classification. This shows our implementation is a rewiring of the usual convolutional networks, with the added benefit of allowing for flexible filter choices. We look forward to seeing how the community uses the `PowerNet` mini-library to incorporate different base matrices for their use-cases.

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
