# OpenReview forum: "PowerNet: Truncated Matrix Power Series as Quasi-Equivariant Layers"
_ICLR.cc/2025/Conference — ICLR 2025 Conference Withdrawn Submission_

### Official Review · Reviewer_v1n6 · 2024-10-31

**Soundness:** 2
**Presentation:** 1
**Contribution:** 1
**Rating:** 1
**Confidence:** 4

**Summary:**

The paper introduces the concept of quasi-equivariance, which relaxes the strict equivariance in deep learning models, and proposes the PowerNet architecture, constructed by interlacing truncated matrix power series with non-linearities.

**Strengths:**

- Overall, the paper is well-written, though some sentences and paragraphs, particularly in Section 3.3, may not have received final polishing. The mathematical formulas are clear and accurate.

- The authors leverage Laurent polynomials and Laurent convolution to construct the proposed PowerNet. While the use of matrix polynomials and Laurent polynomials has appeared in previous machine learning works, to the best of my knowledge, they have not been used in the context of equivariant models.

- Additionally, the authors employ tools from Lie Theory and the Whittaker–Shannon interpolation formula to heuristically demonstrate some theoretical aspects of how PowerNet captures quasi-equivariance.

**Weaknesses:**

- Although the authors mention the introduction of quasi-equivariance in the abstract, they do not provide a clear definition of this property. Figure 1 provides an example of quasi-equivariance, but I still cannot fully grasp what this property entails. It seems to be a form of nearly or almost equivariance.

- The explanation of how the proposed PowerNet layers via Lie Theory and the Whittaker–Shannon interpolation, lead to the quasi-equivariance property, is unclear. I have confirmed that derived formulations are correct, but I lack an understanding of its implications.

- The experimental results are confined to MNIST and CIFAR-10 classification. While I do not expect a wide range of experimental results or that the proposed model needs to be state-of-the-art for any given task, it would be advantageous to include a more varied set of experiments.

Minor typos:
- line 183: exponetiated -> exponentiated
- line 183: Shannon-Whitakker -> Whittaker–Shannon or Shannon-Whittaker

**Questions:**

- Please provide a formal definition of the quasi-equivariance property.

- Offer an intuitive explanation of how the PowerNet architecture embodies quasi-equivariance.

- Include one or two additional experimental results. During the rebuttal phase, I do not recommend that the authors consider conducting a large-scale experiment. I suggest the N-body dataset and MD17 dataset, which are standard for equivariant models.

---

### Official Review · Reviewer_agZ5 · 2024-11-04

**Soundness:** 2
**Presentation:** 1
**Contribution:** 2
**Rating:** 3
**Confidence:** 2

**Summary:**

The authors present PowerNet, a neural network architecture whose core layer computes a matrix-vector product between a truncated Laurent polynomial (based on a chosen base matrix) and an input vector. The base matrix is obtained from generators of relevant Lie algebras.

**Strengths:**

PowerNet is a novel architecture that requires a really small number of parameters.

**Weaknesses:**

* The PowerNet library was mentioned several times, but the code is not submitted and the link to a repository is not provided.
* The paper lacks a theoretical analysis of the model.
* The experiments are restricted to MNIST and CIFAR-10 datasets, the quality of the trained models is not so good. More complex datasets or important domains (like molecular data) are not considered. Furthermore, there is no comparison with existing methods.

**Questions:**

* The number of parameters in a PowerBlock is small. What can you say about the expressive power of the model?
* In contrast to group convolutions, PowerNet uses fully connected layers. Am I right that this architecture might be impractical for high-resolution images?
* Can we find base matrices in a set of structured matrices with, e.g., a fast matrix-vector product?

---

### Official Review · Reviewer_cu7k · 2024-11-04

**Soundness:** 1
**Presentation:** 1
**Contribution:** 1
**Rating:** 3
**Confidence:** 4

**Summary:**

This paper proposes a novel neural network architecture, PowerNet, which draws inspiration from CNNs and GNNs by parameterizing weight matrices as truncated matrix power series. The authors introduce the concept of "quasi-equivariance" and suggest that PowerNet, using Laurent polynomials, can achieve this relaxed form of equivariance. While the concept is intriguing and the architecture shows promise, the paper suffers from a lack of clarity and  rigorous mathematical justification.

**Strengths:**

The paper presents a novel architecture with the potential for efficient learning and flexible structure.
The idea of quasi-equivariance is interesting and could be valuable in handling real-world data imperfections.

**Weaknesses:**

*Lack of Clarity:*
The paper is difficult to read, with the core ideas often obscured by convoluted language and a lack of clear definitions. The connection between weight sharing and quasi-equivariance is not well explained.

*Mathematical Deficiency:*
The authors fail to provide a rigorous mathematical analysis of the relationship between Laurent polynomials and equivariance. There are no proofs or theorems to support their claims.
Specifically, the paper does not clearly explain how the truncation of the power series affects the equivariance properties of the network.
The authors mention using a matrix exponential based on Lie theory, but the implementation and its connection to quasi-equivariance remain unclear.

There is no discussion on the convergence of the power series and under what conditions this convergence might lead to hard (strict) equivariance.

*Limited Experimental Validation:*
While the authors provide some experimental results, these are not sufficient to demonstrate the effectiveness of PowerNet. Comparisons to other state-of-the-art equivariant architectures are missing.

Due to the significant issues with clarity, the lack of mathematical rigor, and the limited experimental validation, I recommend rejection of this paper in its current form. The authors need to:

Improve the clarity and presentation of their ideas, making the paper more accessible to the reader.
Provide a detailed mathematical analysis of the relationship between Laurent polynomials, truncation, and equivariance, including formal proofs and theorems.
Expand the experimental evaluation to include comparisons with other equivariant architectures and explore the applicability of PowerNet to a wider range of tasks.
If the authors can address these weaknesses, the paper may be suitable for resubmission.

**Questions:**

### On Quasi-Equivariance and Laurent Polynomials:

Could you provide a more precise definition of quasi-equivariance? How does it differ mathematically from soft equivariance?

The paper suggests a link between Laurent polynomials and quasi-equivariance. Can you elaborate on this connection with more mathematical detail? Is there a specific property of Laurent polynomials that lends itself to this relaxed form of equivariance?

How does the choice of the base matrix in the Laurent polynomial affect the equivariance properties of the network? Are there any guidelines for selecting base matrices to achieve a desired level of quasi-equivariance for different transformations?

### On Truncation and Convergence:

How is the truncation threshold for the matrix power series determined? Does the truncation affect the degree of quasi-equivariance achieved?

Under what conditions does the truncated matrix power series converge? Does this convergence relate to achieving hard (strict) equivariance? If so, can you provide a mathematical argument or proof?

### On Lie Theory and Implementation:

The paper mentions using a matrix exponential based on Lie theory. Can you provide more details on how this is implemented within the PowerNet architecture? How does this implementation ensure quasi-equivariance?

Can you provide a specific example of how a base matrix would be constructed using Lie theory for a particular transformation (e.g., rotation or scaling)?

###  On Experimental Evaluation:

Can you provide a more comprehensive comparison of PowerNet with other state-of-the-art equivariant architectures, such as those based on group convolution or Steerable CNNs?

How does the computational complexity of PowerNet compare to these other architectures?

Could you explore the application of PowerNet to other tasks beyond image classification, such as image segmentation, object detection, or tasks in other domains like molecular modeling or physics simulations?

---

### Official Review · Reviewer_gjP5 · 2024-11-12

**Soundness:** 2
**Presentation:** 1
**Contribution:** 2
**Rating:** 1
**Confidence:** 2

**Summary:**

The paper proposes a new architecture using a base matrix expnasion which defines a power series. This power series representation has some nice properties, such as quasi-equivariance. Breif experiments show it's potential.

**Strengths:**

This paper proposes an interesting approach.

I like the visualizations.

I like that a software package is introduced.

**Weaknesses:**

This work only appears to work in very simple, shallow 4 layer neural networks. I would like to see this applied to the layers of a VGG or Resnet.

The results are very sparse. Not many baselines.

The paper is a bit confusing to parse.

**Questions:**

Was this approach attempted in traditional neural networks? How did it work there?

---

### Note · Authors · 2024-11-28

**Comment:**

We thank the reviewers for their time and effort.

**Withdrawal Confirmation:**

I have read and agree with the venue's withdrawal policy on behalf of myself and my co-authors.